# The Interplay of Tumor Vessels and Immune Cells Affects Immunotherapy of Glioblastoma

**DOI:** 10.3390/biomedicines10092292

**Published:** 2022-09-15

**Authors:** Mitrajit Ghosh, Anna M. Lenkiewicz, Bozena Kaminska

**Affiliations:** Laboratory of Molecular Neurobiology, Nencki Institute of Experimental Biology of Polish Academy of Sciences, 02-093 Warsaw, Poland

**Keywords:** tumor vasculature, immune suppression, vascular normalization, glioblastoma, immunotherapy, immune checkpoint blockade

## Abstract

Immunotherapies with immune checkpoint inhibitors or adoptive cell transfer have become powerful tools to treat cancer. These treatments act via overcoming or alleviating tumor-induced immunosuppression, thereby enabling effective tumor clearance. Glioblastoma (GBM) represents the most aggressive, primary brain tumor that remains refractory to the benefits of immunotherapy. The immunosuppressive immune tumor microenvironment (TME), genetic and cellular heterogeneity, and disorganized vasculature hinder drug delivery and block effector immune cell trafficking and activation, consequently rendering immunotherapy ineffective. Within the TME, the mutual interactions between tumor, immune and endothelial cells result in the generation of positive feedback loops, which intensify immunosuppression and support tumor progression. We focus here on the role of aberrant tumor vasculature and how it can mediate hypoxia and immunosuppression. We discuss how immune cells use immunosuppressive signaling for tumor progression and contribute to the development of resistance to immunotherapy. Finally, we assess how a positive feedback loop between vascular normalization and immune cells, including myeloid cells, could be targeted by combinatorial therapies with immune checkpoint blockers and sensitize the tumor to immunotherapy.

## 1. Introduction

Glioblastoma (GBM) is the most common and aggressive primary brain tumor in adults. The median survival of glioblastoma patients treated with the available multimodal therapy (encompassing radical surgery, radiation, and chemotherapy) remains less than 15 months after diagnosis, and tumors frequently recur in 6 months [1,2,3,4,5]. Thus, standard treatment options are minimally effective. Surgical resection is not a curative approach and hence is combined with concomitant radiotherapy and temozolomide chemotherapy. Anti-inflammatory steroids such as dexamethasone are given to control peritumoral edema. Radiotherapy/chemotherapy often leads to resistance, iatrogenesis, edema, and immunosuppression associated with lymphocyte depletion and myelotoxicity. The combined temozolomide, radiotherapy, and dexamethasone therapy in GBM patients have been shown to induce immune modulatory effects such as T cell dysfunction, reduced proliferation of T cells, dampened immune responses to the immune checkpoint blockade leading to an increased infection rate, and poor survival [6,7,8,9,10,11]. Thus, these undesirable consequences and immune-related adverse effects lead to poor prognosis and limit therapy outcomes. In addition, unique features of the blood - brain barrier (BBB) and specifically blood—brain tumor barrier (BBTB) in GBM restrict immune cell infiltration and drug influx. Therefore, there is an ardent need for precise targeted therapy and immunotherapy [5,12,13]. Immunotherapy can overcome the resistance of tumor cells to inhibit tumor growth. Furthermore, immunotherapy, in combination with standard treatments of chemotherapy and radiotherapy, can enhance the recognition and elimination of tumor cells by reducing immunosuppression of effector cells, increasing antigenicity of tumor cells, and inducing immunogenic cell death inducers [14,15]. Compelling evidence shows that the composition of the tumor microenvironment (TME) modulates tumor progression and impacts therapy outcomes and the survival of patients [16,17]. GBM is one of the most immunosuppressed tumors, and the massive infiltration and reprogramming of brain resident and peripheral myeloid cells contribute to creating the immunosuppressive and tumor-supporting TME. GBMs developed several mechanisms of immune evasion, which leads to a profound malfunction of the immune response at the tumor site and systemically. GBM is characterized by diffusive tumor growth and the presence of a dense vascular network. The vessels, along with the stromal and immune cells, shape the GBM microenvironment, which can influence disease progression and therapy resistance [18]. Tumor cells require oxygen and nutrients to persist and proliferate, and frequently reside in a close proximity to blood vessels to get near the blood circulation, interacting with other cells for growth and spread. Along with these features, the GBM TME is characterized by intra- and inter-tumor heterogeneity, resulting in the composition of the immune TME being even more complex than in other tumors [19]. GBM cells deploy several strategies resulting in immunosuppression, vessel co-option or hijacking to thrive and constantly reorganize their microenvironment [20]. Therefore, understanding this interplay of vessels and immune cells in the GBM TME is critical for designing an effective therapy. In this review, we discuss the influence of immune cells and vasculature on shaping the GBM microenvironment and how this interacting network could be an important potential target for developing therapeutic strategies that improve immune checkpoint immunotherapy.

## 2. The Aberrant Vasculature in GBM

The tumor vasculature is one of the important components of the TME. The normal structural organization of blood vessels is disrupted in tumors leading to the formation of abnormal vessels which are leaky, collapsed, and disorganized, which contributes to hypoxia, alters tumor metabolism, tumor invasion, immune suppression, and creates specific niches in TME [20,21]. Excessive angiogenesis and fibrosis orchestrate together pathogenic signaling to vascular cells and immune cells [22,23,24]. As such, the tumor vessels are immature and leaky, with dissociated endothelial cells (ECs) and pericytes (PCs), abnormal in shape and spatial distribution. Vessels have abrupted blood flow and are characterized by extravascular accumulation of proteins [23]. This unfavorable TME is characterized by hypoxia, desmoplasia, low pH (acidic), and increased interstitial pressure, which promotes tumor dissemination and survival [4,20]. The GBM vasculature also drives immunosuppression by regulating immune cell function, immunosurveillance, and immune cell trafficking [25].

### 2.1. Structural and Functional Abnormalities of the GBM Vasculature

The abnormal vessels in GBM have structurally aberrant architecture. The structural layers of the BBB are altered and display a fenestration of the basement membrane and endothelium. The tumor vessels have a large diameter and are lined with abnormal ECs and PCs, which are, in turn, connected with an astrocytic end foot and interconnected to neurons and microglia [26,27]. ECs lining the interior of the tumor vessel wall are detached, loosely connected, and have abnormal sprouts. PCs surrounding the ECs and strategically located in between ECs and an astrocyte end foot are loose and display abnormal coverage. As such, the endothelial junctions are not tight, rendering blood vessels leaky [28]. Along with structural deformations, the tumor vasculature shows functional abnormalities. The GBM TME is characterized by hypervascularized vessels to serve increased demands of nutrients and oxygen by rapidly growing tumors. Apart from the nourishment of tumor cells, leading to their survival, hypervascularization promotes GBM invasiveness and progression [25,29]. PCs exhibit less contractility in tumors, which disrupts blood flow and permeability, resulting in brain edema. PC relaxation may facilitate neutrophil transmigration via the loss of PC focal adhesions and reorganization of actin stress fibers. The leakiness of the vasculature disrupts blood flow and governs the dysfunctional homing of lymphocytes. The loss of structural integrity and functional aberrations of tumor vessels induce a collapse of blood vessels under the massive growth and interstitial pressure of the solid mass of the tumor. Furthermore, excessive fibrosis is mediated by cancer-associated fibroblasts (CAFs) that endure forces (by an increase in collagen 1 and hyaluronan production) to compress vessels [23,30,31,32]. Therefore, the tumor vessels show unusual spatiotemporal differences in perfusion [33]. Structural vessel abnormalities coupled with inefficient perfusion led to hampered delivery of therapeutic agents and lymphocytes (Figure 1). The erratic vessels induce hypoxia that further stimulates angiogenesis, and abnormal angiogenesis promotes more hypoxia creating a vicious cycle. These events lead to immunosuppression and reduced trafficking of effector immune cells to TME.

### 2.2. GBM Vasculature as Conduits of the Immune Escape and Therapy Resistance

Tumor cells modulate neovascularization and angiogenesis in several ways, such as sprouting angiogenesis, vasculogenesis, intussusception (splitting a blood vessel), vessel co-option, mosaic vessel formation, and vasculogenic mimicry [4,20]. The hypervascularity results in an imbalanced angiogenic control leading to elevated proangiogenic signaling. The proangiogenic factors may inhibit endothelial–leukocyte interactions and obstruct the intrusion of immune effector cells into TME. The hypoxic and acidic microenvironment formed from distorted tumor vasculature inhibits the infiltration of immune cells such as effector T cells, natural killerNK) cells, antitumor macrophages, and dendritic cells (DCs) [34]. TME hinders the delivery of chemotherapeutics and immunotherapeutics, and supports homing of immunosuppressive immune cells such as neutrophils, T regulatory cells (Tregs) and MDSCs. Tumor ECs produce several angiocrine factors, such as vascular endothelial growth factor (VEGF), fibroblast growth factor (FGF), interlukin-6 (IL-6), interlukin-8 (IL-8), transforming growth factor beta (TGF-β), platelet-derived growth factor-beta (PDGF-β), that foster angiogenesis and immune suppression leading to chemoresistance [35,36,37]. Tumor ECs adopt an anergic state (characterized by a lack of response to proinflammatory stimuli) via suppression of the adhesion molecules linked to leukocyte binding by angiogenic factors. Dysfunctional expression of adhesion molecules, such as intercellular adhesion molecule 1 (ICAM1) and vascular cell adhesion molecule 1 (VCAM1), inhibits T cell adhesion to TME endothelium. Tumor cells upregulate the Endothelin-1, which binds to the Endothelin B receptor on ECs, resulting in the inhibition of ICAM1 expression, which impedes lymphocyte infiltration. Tumor vessel ECs become activated and adopt a prothrombotic, proinflammatory, and cell-adhesive state known as EC dysfunctional activation and facilitate an aggressive phenotype of tumor cells [38]. The tumor vasculature also supports tumor progression by regulating communication with infiltrating immune cells, endothelial cell matrix (ECM) and glioma stem cells (GSCs), which then drives aberrant vascularization [39,40]. GSCs are intrinsically resistant to chemo- and radiotherapy) and contribute to intratumor heterogeneity, invasion, and tumor recurrence [41,42]. Therefore, understanding the underlying mechanisms that render tumor vasculature vulnerable is important for developing effective immunotherapy or enhancement of its effectiveness. An important impediment to the successful delivery of drugs to the brain tumor is the BBTB, which is referred to as a disrupted BBB due to tumor growth compared to the tightly regulated intact BBB in a healthy brain [43]. The BBTB is characterized by aberrant pericyte coverage and loss of astrocytic end feet, rendering a more permeable BBB along with tumor progression. However, this leakier BBTB retains the critical features of the normal BBB, as active efflux transporters are expressed on ECs and tumor cells [44,45]. BBTB is also characterized by heterogenous permeability, and the permeable vessels are responsible for the retention of water and metabolic waste in a neuroparenchymal space leading to edema and increased interstitial and/or intercranial fluid pressure [46]. Though T cell subpopulations and peripheral monocytes can be detected in brain tumors, owing to compromised BBB [47], it limits antigen presentation and effector immune cell infiltration [48,49]. T cell influx is dependent on adhesion and transmigration, but ECs of GBM present little or no adhesion molecules [49], and recent studies showed that the local delivery of chimeric antigen receptor (CAR) T cells induces superior antitumor response compared to the systemic delivery, hinting on limiting the role of BBTB in T cell therapy. Therefore, the identification and optimization of immune cells and drugs that cross this barrier are necessary for designing better antitumor therapies. Strategies to overcome this barrier and deliver drugs and immune cells could be employed. Making use of endogenous influx transport systems such as low-density lipoprotein receptor-related protein 1 (LRP1)/glucose transporter 1 (GLUT1) or overcoming the efflux pumps by using ABC transporter inhibitors could ensure better drug delivery over BBTB [50,51,52]. Homing abilities of stem cells could also be employed for better delivery across BBTB. The use of focused ultrasound, radiation, nanoparticles loaded with drugs, and even manipulation of EC signaling to induce BBB porosity are additional strategies that can be judiciously applied to overcome BBTB and enhance therapeutic delivery in GBM [53,54,55,56].

### 2.3. Contribution of Tumor Vasculature to Local and Systemic Immunosuppression in the GBM Microenvironment

The erratic tumor vasculature comprises tumor-associated blood and lymphatic vasculature. It plays a critical role in the establishment of a local hub that supports immunosuppression, hypoxia, and acidosis, escalates interstitial fluid pressure, and makes a physical barrier to T cell infiltration [57]. The vasculature mediates immune evasion and thwarts T cell-mediated immunosurveillance and antitumor immunity in GBM [57]. Tumor cells release a high amount of VEGF and contribute to angiogenesis via the release of proangiogenic signaling molecules, such as placental growth factor (PGF), VEGF-C, VEGF-D, and PDGF-C [58]. Leaky vessels stimulate hypoxic and acidic TME and drive angiogenic signaling that promotes immunosuppression through several mechanisms [31,59,60]. VEGF blocks cytotoxic T cell infiltration and activity by modulating the inhibitory checkpoints of T cells. Cytotoxic T lymphocyte-associated molecule-4 (CTLA-4), programmed cell death receptor-1 (PD-1), and programmed cell death ligand-1 (PD-L1) are the most known and well-studied components of inhibitory checkpoint pathways. Anti-CTLA-4 (ipilimumab) and anti-PD-1 (nivolumab) antibodies have been approved by the Food and Drug Administration (FDA) for the treatment of advanced melanoma [61]. CAR T-cell therapy, based on reprogramming patient T cells and expression of a synthetic receptor that binds to the tumor antigen, has been a significant development in personalized cancer treatment [25,62]. VEGF upregulates T cell immunoglobulin mucin receptor 3 (TIM3) and lymphocyte activation gene 3 (LAG3) protein expression, contributing to T cell exhaustion [58,63]. VEGF restricts T cell activation by the inhibition of DCs maturation and antigen presentation. VEGF attracts immunosuppressive cells such as Tregs, MDSCs, and protumor M2-like (glioma-associated microglia and macrophages (GAMs) [64,65]. Fibrosis or desmoplasia is uncontrolled in tumors. Hypoxia stimulates fibroblasts through connective tissue growth factor (CTGF), TGF-β, and sonic hedgehog (Shh) signaling. Moreover, fibroblasts generate elevated solid stress and tissue stiffness that compress the blood and lymphatic vessels, elevating the interstitial fluid pressure and reducing blood flow. The events may accelerate the invasiveness of tumor cells [20,23,66]. Immunosuppression in the GBM TME is the prime reason for the failure of immunotherapy. The aberrant vasculature, along with tumor cells, glioma-stem-like cells, resident (microglia), and peripheral myeloid cells via immunomodulatory factors such as: TGF-β, migration inhibitory factor (MIF), interleukin 6 and 10 (IL-6, IL-10), prostaglandin E-2 (PGE-2), and surface ligands: PD-LI, lead ultimately to T cell exhaustion and anergy. Infiltration of Tregs that blunt the antitumor T cell response, loss or reduced expression of major histocompatibility complex (MHC II) by antigen-presenting cells (APCs) such as microglia, M2-like GAMs, and infiltrating neutrophils expressing arginase-1, cooperate to induce severe immunosuppression [67,68]. CD95 (Fas/APO-1), a death receptor family member that regulates tissue homeostasis of the immune system by inducing apoptosis, has been implicated in tumorigenicity in multiple cancers, including GBM. Induction of survival and migration of glioblastoma cells by the CD95/CD95L system has also been described [69,70]. However, not just tumor cells but other cells of TME can respond to the CD95 pathway. Though not assessed in GBM, tumor cells of human breast, renal, colon, and other cancers displayed expression of CD95L in tumor-associated ECs. The ECs then form a unique death barrier that selectively kills cytotoxic T cells but allows an entry of T regs, creating an immunosuppressive “cold” tumor [71]. Using paracrine signaling from VEGFA, IL-10, and PGE2, this tumor endothelial death barrier is established, and inhibition of VEGFA and PGE2 inhibits FasL expression leading to tumor attenuation mediated by cytotoxic T cells [71]. Similarly, CD95L-mediated killing of tumor-infiltrating cytotoxic T cells as a part of a “tumor counterattack model” [72] has been suggested to operate in gliomas as well. The FasL expressed on the GBM tumor cell surface binds to Fas on T cells, leading to apoptosis of T cells, thereby enabling the killing of Fas+ T cells and eventually evading lysis by T cells [73].

## 3. The Immune Landscape in GBM

Under physiological conditions, neurons, astrocytes, microglia, oligodendrocytes, and endothelial cells interact with each other to coordinate the proper functioning of the brain. Recent single-cell RNA sequencing studies in mice and humans, along with cell fate mapping and cell tracing experiments, revealed the presence of diverse types of immune cells in a healthy brain, experimental gliomas, and GBMs [68,74]. Intra-parenchymal microglia, the central nervous system (CNS) border-associated macrophages (BAMs) consisting of perivascular, choroid plexus and meningeal macrophages, DCs, NK cells, neutrophils, eosinophils, and mast cells are present in the CNS under physiological conditions [75]. Under pathological conditions, tumor cells release numerous factors activating endothelial cells and attracting numerous immune cells from the periphery.

### 3.1. Multifaceted Action of Glioma-Associated Myeloid Cells on TME and Antitumor Immunity

Glioma-associated microglia and macrophages (GAMs) consist of up to 30% of the tumor mass and are determinants of tumor progression [76]. Recent multi-omics studies demonstrated functional plasticity, which allows myeloid cells to adapt to the prevailing conditions in the TME of a specific tumor [77]. There are many molecular mechanisms responsible for the regulation of GAM activity and functions within TME. Tumor-derived factors such as colony-stimulating factor 1 (CSF1; also known as M-CSF), granulocyte-macrophage colony-stimulating factor (GM-CSF), CC-chemokine ligand 2 (CCL2), CCL7 (or MCP-3), GDNF, IL-33, CXCL12 (SDF-1) and EGF are responsible for myeloid cell recruitment and promote tumor growth [78,79,80,81]. However, a significant impact on the polarization of myeloid cells is exerted by hypoxia, nutrient availability, and lymphocyte-derived factors. Proinflammatory M1-like macrophages were detected within normoxic tumor regions, while protumor M2-like macrophages were in hypoxic regions [82]. GAMs-derived IL-1β induces activation of the p38 MAPK pathway and CCL2 production, leading to GSC proliferation. TGF-β-dependent secretion of matrix metalloproteinases 9 and 2 (MMP-9, MMP-2) by ECM disruption upregulates tumor invasion. The migratory capacity of GBM cells is enhanced by the upregulation of PDGF receptor beta (PDGFR-β) by GAMs, stress-inducible protein 1 (STI1), and EGF [83]. Signal regulatory protein alpha (SIRPα) and inhibitory receptor Ig-like transcript 2 (ILT2), which are expressed on GAMs, inhibit phagocytosis of tumor cells.

GAMs-mediated immunosuppression involves several mechanisms. Production of PGE_2_ by GAMs inhibits the secretion of IFN-γ and cytolytic activity of NK cells and blocks differentiation of Th1 into effector T cells as well as expression of IL-2 and IL-2 R on effector T cells. Moreover, GAMs-dependent PGE_2_ production has a negative effect on the activity of DCs by disturbing the secretion of IL-12 in these cells, simultaneously promoting differentiation and accumulation of Tregs, expansion, and suppressive activity of MDSCs. Production of cytokines and chemokines such as CCL2, CCL5, CCL20, CCL22, IL-10, and TGF-β by GAMs accelerates Tregs recruitment and expansion. Moreover, IL-10 and TGF-β inhibit activation and differentiation of effector T cells, proliferation, and cytotoxicity of NK cells and promote the immunosuppressive phenotype of DCs by inhibition of IL-12 secretion and costimulatory molecule expression. GAMs via the production of kynurenine and its derivatives or by depletion of L-tryptophan in TME inhibit proliferation and activity of NK and T cells. A separate mechanism of GAMs-dependent immunosuppression involves a direct engagement of immune cell inhibitory receptors. Non-classical HLA class I molecules E and G expressed on GAMs interact with NK and CD8+ cells inhibiting their proliferation and cytotoxic activity, which in consequence, leads to exhaustion of these populations. Moreover, HLA-G-ILT2 interactions induce tolerogenic DCs, inhibit proliferation of CD4+ T cells and B cells, and promote Tregs differentiation. Compelling evidence clearly shows that the extensive adaptive mechanisms and plasticity of glioma-associated microglia and macrophages, together with the signals provided by gliomas and TME, result in creating a unique immune TME in GBMs.

### 3.2. T Cells, NK Cells, and DCs in GBM

A successful antitumor response requires the recruitment of immune cells to the tumor, where they can execute their effector functions. Several populations of tumor-infiltrating lymphocytes (TILs), such as CD4+ T helper (Th), CD8+ T cytotoxic (Tcyt), and regulatory CD4+CD25+FoxP3+ T-cells (Tregs) are found in the GBM TME [84]. Depletion of cytotoxic CD8+ T cells in the GBM TME may result from persistent antigen exposure, chronic TCR signaling or changes in gene expression, and accumulation of immunosuppressive metabolites and cytokines secreted by GAMs. Increased and sustained expression of many inhibitory receptors, including the immune checkpoint proteins mediated by MDSCs and Tregs [85], and increased tolerance induced by recruitment and expansion of Tregs contribute to the suppressive effect. Moreover, sequestration of T cells in the bone marrow via the Sphingosine-1-phosphate receptor 1 (S1PR1 or S1P1) has been proposed a novel mechanism of effector T cell immunosuppression [86]. The percentage of CD4+ infiltrating T cells correlates with a glioma grade, and leaky arteries may facilitate the transmigration of T cells. While transmigration of T cells is mediated by GBM endothelial cells, factors secreted by GBM and GAMs, such as TGF-β and IL-10, are responsible for the lack of adequate activation of T cells. Tumor-derived CD4+ effector memory T cells expressed high levels of PD-1 and were functionally exhausted after interaction with-MDSCs [87,88].

Immunosuppressive T cells (Tregs) are the main mediators of immunological tolerance due to their inhibitory effect on the immune response. GBM cells and GAMs contribute to the recruitment and maintenance of Tregs in the TME by secreting soluble molecules such as CCL22 [89,90] and the tryptophan (Trp) metabolizing enzyme, indoleamine 2,3-dioxygenase 1 (IDO). Tregs suppress effector lymphocytes through a number of mechanisms. First, Tregs are highly dependent on IL-2; thus, competition restricts the availability of this cytokine to T cells. Second, Tregs via cytotoxic T lymphocyte antigen 4 (CTLA-4) transmit suppressive signals to APCs, thus reducing their maturation and capacity to prime and/or activate T cells. Moreover, Tregs-mediated production of immunosuppressive cytokines (such as IL-10, IL-35, and TGF-β), granzymes, and perforins downregulates the activity of APCs and effector T cells or kill them, respectively [91]. Finally, conversion of ATP into immunomodulatory metabolite adenosine via CD73 and CD39t can prevent optimal T cell activation by the engagement of adenosine A2A receptor [92]

NK cells are innate lymphoid cells with cytotoxic activities against “non-self” target cells, pathogens, and tumor cells. These cells exert immuno-modulatory functions through several mechanisms: production of cytokines and enzymes such as perforin and granzymes, by crosstalk with monocyte/macrophages, DCs, B and T lymphocytes as well as via interactions of cell death receptor and/or antibody-dependent cellular cytotoxicity during the antitumor response against GBM [93,94]. TME-mediated signals, i.e., PGE_2_, can affect NK cells reducing their cytolytic activity and ability to secrete INF-γ [95]. Increased levels of IDO in GAMs result in impaired proliferation and activity of innate lymphoid cells, while IL-10 and TGF-β downregulate stimulatory receptors NKG2D-NKp30, thus inhibiting cytotoxic activities of NK cells.

DCs, as professional APCs, are responsible for capturing, processing, and presenting antigens to T and B cells in lymphoid organs, and they are rare inside the brain parenchyma under homeostatic conditions [96]. Bone-marrow-derived precursors are differentiated into DCs by Flt3L (fms-like tyrosine kinase 3 ligand) or GM-CSF. GM-CSF increases the frequency of Th1 and helper T cell 2 (Th2) cells, while Flt3L mainly increases Th1 cell frequency. In GBM patients, DCs are recruited to the TME via afferent lymphatic vessels or endothelial venules and can support antitumor immunity in an immunostimulatory or immunosuppressive manner. Mature DCs cross-present tumor antigens to CD8+ T cells, and they can boost the antitumor activity of NK cells and increase Th1 stimulation [97,98]. Mature DCs can induce deletion, anergy, or downregulation of Tregs activities, at the same time supporting activation, proliferation, and differentiation of cytotoxic and helper T cells [85,99]. In the GBM TME DCs are reprogrammed by TME to induce Tregs activity and suppress the proliferation of cytotoxic T and NK cells. The presence of immunosuppressive signals IL-10 or IL-27, insufficiency of costimulatory molecule B7-1 expression, or secretion of IL-12 can induce tolerogenic DCs. DC-derived immunoglobulin receptor 2 (DIgR2) and Notch ligands directly suppress T cells. Cytokines present in TME suppress DCs maturation resulting in reduced effector T cell activation [85,100]. Moreover, MDSC-derived oxidized lipids accumulating in the DCs inhibit their capacity to present tumor antigens [101,102,103]. Interactions of TME-generated fibrinogen-like protein 2 (FGL2) with GM-CSF signaling inhibit activation of DCs [103]. DCs maturation can also be blocked in a Nrf2-dependent manner. GBM cells increase Nrf2 expression in DCs, and its inhibition results in increased DCs maturation and T cell activation [100].

The presented immune cell dysfunctions contribute to the lack or scarcity of functional tumor-infiltrating cytotoxic CD8+ T lymphocytes (CTLs) in the GBM TME, so GBMs are considered immunologically “cold” tumors. In contrast to CTLs, Tregs infiltrate the GBM TME and, together with tumor-supportive microglia and macrophages, constitute the predominant suppressive cell populations within GBMs, which make them less responsive to immunotherapies [104].

### 3.3. Failure of the Immune Checkpoint Blockade in GBM

Current immunotherapy includes immune checkpoint blockade (ICB) with specific antibodies (several ongoing trials such as CheckMate 143 phase III, CheckMate 498 phase III, CheckMate 548 phase III), therapeutic vaccines (including peptide vaccine, DC vaccine, DNA vaccine trials) (such as ACT IV phase III, NCT00045968 phase III), engineered CAR T cell therapy (several ongoing trials), adoptive cell therapy, monoclonal antibodies and oncolytic viruses (several ongoing trials) [105,106,107,108,109]. However, these strategies have not been very effective in GBM patients, and no major breakthrough for GBM treatment or patient survival has been achieved [109,110,111]. Immune checkpoints successfully targeted in tumor immunotherapy are PD-1, PD-L1, and CTLA-4. Monoclonal antibodies targeting PD-1 protein and its ligand PD-L1 to enhance the cytotoxic activity of CD8+ T cells have been successful in melanoma and non-small cell lung cancer [112]. The first randomized phase III clinical trial CheckMate 143 (NCT02017717) with the anti-PD-1 monoclonal antibody, nivolumab, in patients with recurrent GBM failed to prolong overall survival in comparison to bevacizumab (anti-VEGF), and the study was subsequently closed [113]. The response rate to ICB immunotherapy was not significant, with less than 15% of cancer patients showing slight improvements. To date, no phase III clinical trial has shown success in GBM patients [107,108]. However, a recent open-label pilot study showed that treatment with anti-PD-1 (programmed cell death protein 1) antibody (pembrolizumab) prior to tumor resection escalated both local and systemic antitumor immune responses in recurrent GBMs, and patients had significantly extended overall survival [114]. One of the main reasons for those failures is that ICBs reduce the immunosuppression exerted on T cells, stimulate the immune response against a specific antigen or activate DCs responses against the tumor, but those mechanisms do not operate in the “cold” GBM TME. Concomitant targeting of both innate and adaptive immunity and a combination of multiple immunotherapies hold promise for better efficacy of ICB in clinical studies.

## 4. Reprograming of the GBM TME to Improve ICB Immunotherapy

GBMs are composed of highly proliferative tumor cells with low mutational burden and neoantigen levels, limiting the ability of CTLs to recognize tumor cells and initiate an antitumor response, even after CD8 T cells are re-activated with ICB. A comprehensive immunogenomic analysis of >10,000 tumors comprising 33 diverse cancer types by TCGA demonstrated that the composition of the immune TME impacts the survival of patients and revealed differences in TME composition [115]. In order to increase the efficacy of ICB, the “cold” GBM TME has to be tamed and converted to “hot” TME by enhancing infiltration and activity effector immune cells. Though most immunotherapeutic approaches are designed to enhance the antitumor activity of CTLs, the concomitant response of myeloid cells can substantially influence treatment outcomes. Innate and adaptive immune cell activities are closely linked during tumor progression and in response to therapy. Strategies to ablate myeloid populations are not favored because the immune cells maintain immune homeostasis; DCs and antitumor myeloid cells play a key role in mounting an effective antitumor immunity, either directly or by enabling CTL function. Thus, reprogramming myeloid cells to an immunostimulatory state is an attractive approach to augmenting immunotherapy outcomes. Numerous signaling pathways in myeloid cells could be targeted to promote antitumor immunity: CSF-1/R axis, CCL2, CD40, and IL-6. CSF-1/R antibody therapy sensitizes tumors to ICB by reprogramming myeloid cells to support the influx of T cells and re-activate exhausted CTLs [116]. Selective reprogramming of Tregs to immunostimulatory states by inhibiting their suppressive functions can augment immunotherapy outcomes while maintaining systemic Tregs. Preclinical studies have found that reprogrammed Tregs adopt the immunostimulatory effector T cell phenotype. Exploiting the CD25, 0X40, GITR, 4-1BB, and IDO signaling pathways reduced immunosuppression by these cells in preclinical studies and showed feasibility in clinical trials [117,118].

## 5. Tumor Vessel Normalization to Improve ICB Immunotherapy in GBM

Strong evidence shows that vasculature plays a critical role at the interface between tumor cells and immune TME in the regulation of tumor growth and immunotherapy efficacy [57]. This conceptualization triggered the idea of vascular normalization [119]. Though originally antiangiogenic agents were used to starve the tumor, thus halting tumor progression and improving patient survival, they failed to provide significant survival benefits in clinical studies. The anti-VEGF antibody bevacizumab, which blocks the VEGF/VEGFR-dependent survival and growth of vasculature, did not elicit the expected results in cancer patients. Further, preclinical studies demonstrated that vessel pruning leads to increased hypoxia, which supports tumor growth and metastatic dissemination [23]. The concept of vascular normalization or healing aims at restoring a stable phenotype with more fortified and mature vessels with the appropriate pericyte coverage. Such normalized vessels could promote the accumulation, penetration, and antitumor activity of immune effector cells, while reducing hypoxia can convert the immunosuppressive TME into an immunostimulatory one [120]. In line with this concept, several clinical trials are ongoing to test combinations of antiangiogenic compounds with ICB-based immunotherapies [21,121,122,123,124]. To date, vascular normalization (improvement of perfusion in hyperpermeable vessels) encompasses stroma normalization, endothelial reprogramming, and combinatorial treatments to enhance the efficacy of immunotherapy [20,125,126,127,128]. Compromised tumor perfusion is attributed to the abnormal and hyperpermeable vessels in the tumor. The increase in perfusion has improved ICB response. Inhibition of angiotensin or CXCL12/CXCR4 signaling, targeting extracellular collagen, hyaluron, and CAFs, improves perfusion and decompression of tumor vessels and has a positive impact on ICB [129,130]. With the right dose of anti-VEGF agents, vessel tortuosity and hyperpermeability can be reduced, leading to improvements in tumor vessel perfusion, oxygenation, and drug delivery potential. Anti-VEGF agents could also increase pericyte coverage on tumor vessels and strengthen immature vessels. Better tumor perfusion via vessel normalization improves the tumoricidal efficacy of immune cells, which can also kill more resistant tumor stem-like cells [127,131]

Genetic depletion of regulator of G-protein signaling 5 (Rgs5) expression in mice promoted conversion of immature PDGFR-β+ PCs to mature alpha-smooth muscle actin positive and neuro-glial antigen 2 positive (αSMA+, NG2+) PCs, but their overall coverage of the vasculature was unaffected. This phenotypic change in tumor vessels led to a reduction of hypoxia and vascular leakiness in the tumor, paving the way for effector immune cells to enter tumor parenchyma [132]. Vessel normalization results in an increased and more uniform distribution of adhesion molecules on the luminal surface of ECs lining the tumor blood vessels, thus allowing more efficient immune cell infiltration into tumors. Dual Ang2 and VEGF blockade induced vascular normalization resulting in lymphocyte priming by APCs, GAM polarization to the M1-like phenotype, and homing of activated, IFNγ-expressing CD8+ T cells within the perivascular space [21,133]. This dual blockade upregulated PD-L1 expression in tumor endothelial cells through IFNγ. Therefore, these studies demonstrate an intricate relationship between immune cells and tumor blood vessels and provide a strong rationale for combining antiangiogenic therapy with immunotherapy.

Activated eosinophils can also promote the normalization of tumor vessels. The exact mechanism is unknown, but they act through the polarization of GAMs into the M1-like phenotype through eosinophil-derived IFNγ and tumor necrosis factor (TNF) signaling, which can decrease VEGF production. The normalized blood vessels improve T cell infiltration, which results in a positive feedback loop that supports more M1-like myeloid cell polarization and VCAM1 expression on ECs for efficient T cell influx. Enhanced effector T cell infiltration and activity (e.g., via formation of high-endothelial venules) [134] reduced infiltration and suppressive functions of Tregs and MDSCs, and increased presence and activity of APCs (e.g., DCs) have been described [134]. Together, these studies confirm that vascular normalization has an immune stimulatory role by enhancing antitumor immunity. The reciprocal regulation of immune cells and tumor vasculature under combined antiangiogenic (under low dose) and immunotherapeutic approaches is gaining more attention [22,34]. The observed interplay between tumor blood vessel remodeling and immune TME reprogramming has led to studies that examined the effect of vascular normalization on immune checkpoint blockade and vice versa [34].

## 6. Perspectives and Further Directions

Vascular normalization and reprogramming of immune cells in TME, when combined with immune checkpoint inhibitors, hold a strong potential for the improvement of GBM therapy. Immunotherapies such as ICB, oncolytic viral vaccines, and immunostimulatory therapies using STING agonists suppress tumor angiogenesis, often through IFNγ, GAMs, and CD8+ T cells. Understanding how specific immune cells regulate various aspects of vascular normalization, including vessel pruning, pericyte recruitment, alleviation of hypoxia, and oxygenation, must be explored to derive complete benefits. A combination of judicious dosing of antiangiogenic drugs with vascular decompression and stromal and vascular normalization, along with tumor immune reprogramming of Tregs or myeloid cells, may provide benefits for immunotherapy (Figure 2). Bevacizumab is often used to inhibit VEGF and attenuate tumor spread by preventing tumor angiogenesis. However, the dosing needs to be rationally designed as though higher doses are well tolerated, and they give rise to several other associated toxicities. Hypertension, thromboembolic events, gastrointestinal perforation, cerebral bleeding, wound healing complications, and decline in neurocognitive functions are some of its side effects [126,135,136,137]. Therefore, it is of utmost importance to judiciously modulate the anti-angiogenesis dosing to combine with immunotherapy and derive maximum benefits and not elicit other alternative therapy resistance pathways or immune-related adverse events [126]. A low dose of bevacizumab combined with immunotherapeutic strategies would normalize vessels, prevent desmoplasia and ensure effective immune cell interactions. CAR T cells and adoptive T cell transfer can induce compensatory immunosuppression in the brain with the influx of Treg cells [138], while vaccination strategies could lead to the tumor immune escape [139]. Thus, ICBs, CAR T cells, and vaccination can benefit if immunosuppression is limited/removed, specific localized delivery is achieved, and cellular heterogeneity is effectively dealt with [140]. Combining low-dose antiangiogenic drugs with immunotherapy or combining several immunotherapies could be an answer to dealing with the TME of GBM. Thus, synergizing these modalities to promote antitumor effects by coupling immune-vascular interactions is critical. Combinations can promote enhanced effector T cell influx and activity (via formation of high-endothelial venules), reduce effector T cell dysfunction, attenuate the presence and suppressive functions of Tregs and MDSCs, and support the presence and activity of APCs such as DCs [135,141]. Modulation of myeloid cells mediated signaling and their interactions with stomal cell partners (astrocytes, oligodendrocytes, pericytes, fibroblasts, ECs), along with the induction of long-lasting normalization and appropriate oxygenation, may pave the way to improvement in survival of GBM patients [13,15,142,143]. Development of biomarkers better characterizing TME of GBM with specific genetic alterations, proper visualization, or new markers of vascular normalization are prerequisites for predicting immunotherapeutic success. The lack of validated biomarkers and strategies for monitoring therapy outcomes are major challenges in GBM therapies. These issues are augmented in immunotherapy settings in GBM, where one need to distinguish between pseudo-progression (due to tumor immune infiltration) vs. true progression of tumor volume, immune checkpoint inhibition, and edema. Hence, identifying the spectrum of responsiveness is important [108]. Therefore, combination therapies hold the keys to the success of immunotherapy by eliciting immunostimulatory responses in GBM and transforming a “cold” tumor into a “hot” one. A combination of immunotherapy with standard treatment of chemotherapy/radiotherapy can benefit from immune augmenting effects. For example, temozolomide or radiotherapy that creates lymphodepletion can be used to improve the efficacy of CAR T cell therapy and survival. Furthermore, metronomic dosing of temozolomide induces immunostimulatory macrophages in TME. It has also been shown that the combined temozolomide and immunotherapy increased CD8+ T cell infiltrates and reduced MDSCs [107,144,145]. Several studies demonstrate that combining chemotherapy and immunotherapy modulates the immune system to achieve effective treatment benefits; for example, ongoing clinical trials such as NCT0357661, NCT00626015 (after surgery) or NCT02010606 (with recurrent GBM) will show the validity of this promising approach [108,146,147,148,149]. Along with these, targeting multiple arms of immunotherapy and combining multiple immunotherapies could also provide key information and survival benefits. Therefore, in the future, the combination of immunotherapy with standard therapies or mixing immunotherapies will hopefully create synergistic antitumor activities. One of the most important therapeutic targets, in this context, would be the positive feedback loop between activated immune cells and normalized vessels. Using combinational strategies in preclinical and clinical studies would result in the enhancement of immunotherapy efficacy.

## Figures and Tables

**Figure 1 biomedicines-10-02292-f001:**
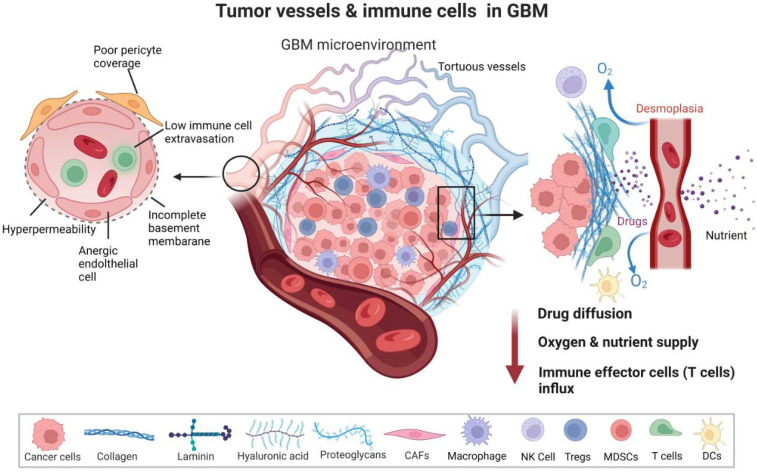
Tumor vessels and immune cells in Glioblastoma (GBM). The tumor microenvironment of GBM is characterized by erratic, tortuous vessels that have anergic endothelial cells, poor pericyte coverage and incomplete basal membrane. The vessels are therefore hyperpermeable and leaky, subsequently restricting immune cell extravasation. There is intricate relation of blood vessels with recruitment of immune cells to tumor; as the extra cellular matrix components (collagen, laminin, hyaluronic acid, proteoglycans) of tumor and desmoplastic tumor vessels orchestrate a barrier for effector immune cells influx, drug delivery and mediated with reduced oxygen and nutrient supply. T effector cells, natural killer (NK) cells, dendritic cells (DCs) are restricted from entry to tumor niche, which is replete with T regulatory cells (Tregs), myeloid derived suppressor cells (MDSCs), cancer-associated fibroblasts CAFs-, and glioma associated M2 macrophages. *Circle* area of the blood vessel is projected to details of the vessel. *Rectangle* area depicted in details the barrier (of vessels and endothelial cell matrix) where drugs, immune cells and oxygen supply is hindered to the tumor cell, and hence *red* arrow shows decrease in these parameters at tumor core. Created with BioRender.com.

**Figure 2 biomedicines-10-02292-f002:**
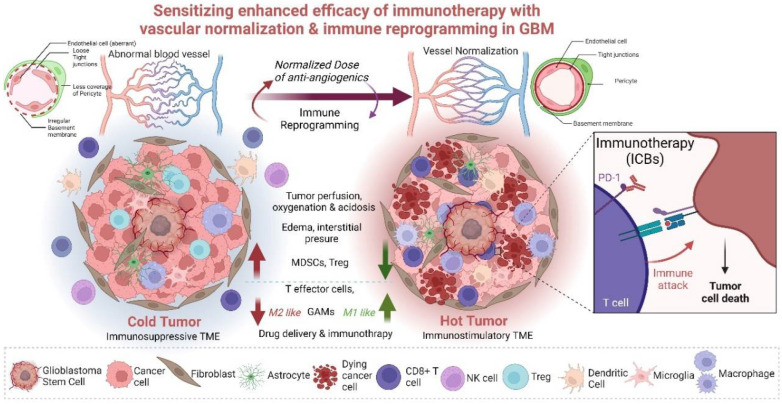
Enhancing immunotherapy in Glioblastoma (GBM) with the reciprocal regulation of the vascular normalization and immune reprogramming. The “cold”, immunosuppressive tumor microenvironment (TME) in GBM is depicted with no effector T cells, dendritic cells (DCs) and natural killer (NK) cells but with immunosuppressive myeloid derived suppressor cells (MDSCs), M2-like glioma associated macrophages (GAMs) and T regulatory cells (Tregs). The “cold” tumor TME is characterized with abnormal tumor blood vessels that are tortuous and leaky due to irregular basement membrane, aberrant endothelial cells (ECs) with loose tight junctions, and immature and irregular pericyte coverage. These events lead to less perfusion and oxygenation (hypoxia) which increases an acidic microenvironment, consequently leading to increased edema, interstitial pressure, less drug delivery and unsuccessful immunotherapy. Using a judicious (normalized) dose of anti-angiogenics, vessel normalization can be attained where the basement membrane is regular, with proper ECs, tight junctions and matured pericyte coverage that fortify the vessels leading to increased vessel perfusion, oxygenation and reducing acidosis. Less edema and interstitial pressure create a conducive TME for homing effector T cells and creating “hot” immunostimulatory TME that homes DCs, NK cells and no or less Tregs and MDSCs. M1-like macrophages are present along with microglia, and other stromal cells. This reciprocal regulation of vascular normalization leading to immune reprogramming and vice versa sensitizes the “hot” tumor for effective immunotherapy. The *purple* large arrow shows that by reciprocal regulation of immune reprogramming and normalized dose of anti-angiogenics can induce vessel normalization from abnormal vessel architecture. The *red* arrows depict immune characteristics of “cold tumor with increased MDSCs and Tregs and less T effector cells and more M2 like GAMs leading to increased edema, interstitial pressure, acidosis, and less tumor perfusion, oxygenation, drug delivery and less affective immunotherapy. The *green* arrows depict immune characteristics of “hot” tumor with less MDSCs and Tregs and more T effector cells and more M1 like GAMs leading to decreased edema, interstitial pressure, acidosis, and more tumor perfusion, oxygenation, drug delivery and more affective immunotherapy. Created with BioRender.com.

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
