# Peer review of "The Interplay of Tumor Vessels and Immune Cells Affects Immunotherapy of Glioblastoma"

_biomedicines, 2022, doi:10.3390/biomedicines10092292_

Round 1
Reviewer 1 Report
It is often forgotten how much vasculature can be decisive in inducing immunosuppression. In this review, the authors discuss how the abnormal tumor vasculature can elicit immune suppression in the tumor microenvironment. They focus on the functional alteration of local immune infiltrating cells and increasing immunosuppression, and how different subset cells within the tumor microenvironment, MDSCs, and GAMs, are involved in these mechanisms.
Furthermore, the authors discuss the vasculature impact on the failure of the checkpoint inhibitors in GBM patients and bring original insight into the therapy of GBM, such deadly cancer.
There are some minor issues.
1. The authors should address:
- the concept of the BBTB (blood-brain tumor barrier);
- the BBTB potential impact on immune cell infiltration;
- some potential strategies to overcome the BBTB.
2. Although not directly addressed in the GBM, the author should mention the "tumor endothelial death barrier hypothesis", considered one of the causes for which some tumors are cold. The FasL expressed by endothelial cells can inhibit the killing ability of Fas+ activated CD8 T cells.
3. Bevacizumab can improve the performance of checkpoint inhibitors, as described in the review. It should be intriguing to consider the effects of Bevacizumab in combination with other immunotherapeutic strategies considering the previously demonstrated effects, such as i. TME reprogramming; ii. antitumor immunity restoration; iii. recruitment of reactive infiltrating T cells.
Reviewer 2 Report
In this review, authors present all characteristics of vasculature and immune cells from glioblastoma micro-environment. On the opposite to melanoma, this type of tumor is particularly immune-resistant intrinsically and by adaptation. Therefore, the challenge is now to better know all parameters and cellular partners involved in this resistance to optimize immunotherapy of glioblastoma.
By all the information and details, this review contributes to a better knowledge of glioblastoma TME and suggest some approaches to improve immunotherapy. However, it is important in the introduction to better detail the current conventional treatment of glioblastoma. The first reason is to show the gap between the current therapy and the immunotherapy proposed in this review. The second reason is because classical treatment of glioblastoma involves chemotherapy/radiotherapy inducing apoptosis and this mode of action should be synergistic with a reactivated immune system. Therefore, combinatorial treatment with classical chemotherapy + immunotherapy might be the future and should be clearly mentioned in the perspectives (do not consider combination of angiogenic- and immuno-modulators).
Another remark is that, surprisingly, several reviews were published during the last years about immunotherapy of glioblastoma and none of them are mentioned in references (Lim et al, 2018, Jackson et al., 2019, Medikonda et al., 2020, Yu & Quail, 2021). The “old” review of Lim, for example, could be cited at least to refer to all the clinical trials initiated for treating glioblastoma by modulating immune system. Authors could enrich their publication by references that could complete some domains that they cannot develop exhaustively, especially the therapeutic axe (the tumor biology is very well treated).
